Turkish validation of a resilience scale from older people’s perspectives: resilience scale for older adults

Palanbek Yavaş Seher 1
http://orcid.org/0000-0002-7675-1391 Baysan Caner 2 caner.baysan@ege.edu.tr
1 Department of Public Health, Canakkale (18th March) University , Çanakkale , Turkey
2 Department of Public Health, Ege University , İzmir , Turkey
Albuquerque Ulysses
Electronic publication date: 2025 Jan 21
Publication date: 2025
Volume: 13
Electronic Location ID: e18837
Received 2024 Oct 11; Accepted 2024 Dec 18
Copyright: © 2025 Palanbek Yavaş and Baysan
Copyright year: 2025
Copyright holder: Palanbek Yavaş and Baysan
License: This is an open access article distributed under the terms of the Creative Commons Attribution License, which permits unrestricted use, distribution, reproduction and adaptation in any medium and for any purpose provided that it is properly attributed. For attribution, the original author(s), title, publication source (PeerJ) and either DOI or URL of the article must be cited.
License URL: https://creativecommons.org/licenses/by/4.0/

Keywords: Aged, Reliability and validity, Resilience, Elderly

Funding: The authors received no funding for this work.

==============================
Background

As the global population ages and life expectancy increases, older adults encounter challenges like chronic illnesses and losing loved ones; resilience is crucial for adapting to these difficulties. This study aims to culturally and linguistically adapt the psychological resilience scale designed for older adults to the Turkish context.

Methods

This methodological study included 566 individuals aged 65 and older. All participants completed the Resilience Scale for Older Adults, the Perceived Stress Scale, the Geriatric Depression Scale, and the Quality of Life in Older Adults Scale—Short Form. Construct validity was assessed through confirmatory and exploratory factor analyses, while convergent and discriminant validity were evaluated through the correlation of the scales. Reliability was determined using internal consistency and the split-half method.

Results

The scale retains its original structure with 33 items and four sub-dimensions (Intrapersonal, Interpersonal, Spiritual, and Experiential). The content validity index of the scale is 0.98, with item loadings ranging from 0.406 to 0.947, according to exploratory factor analysis. Confirmatory factor analysis indicates good fit indices. Convergent validity is supported by a strong positive correlation (r = 0.657, p < 0.001) between the Resilience Scale for Older Adults and the Quality of Life in Older Adults Scale—Short Form. Reliability measures include a Cronbach’s alpha internal consistency coefficient of 0.93 and a Guttman split-half reliability coefficient of 0.723.

Conclusion

The results show that the Turkish version of the Resilience Scale for Older Adults is a valid and reliable measurement tool.

Introduction

People face many unfortunate events throughout their lives. These unfortunate events can vary widely; for example, disasters such as wars, floods, and earthquakes; all forms of violence, including rape and child abuse; traffic accidents; the loss of a parent; challenging disease processes; and so on. Traumatic events, uncertainties or difficulties can instantly turn people’s lives upside down. However, individuals respond differently to these situations. It has been observed that some people develop post-traumatic stress disorder, others develop the belief that the world is a dangerous place, and some overcome difficulties and continue their lives in a meaningful and productive manner (Southwick, Charney & DePierro, 2023). At this point, the concept of psychological resilience has been defined by many authors. In the years when the concept first emerged, Kobasa (1979) defined it as the ability to see negative experiences from a positive perspective, Murphy (1987) as the ability to adapt to difficult situations, and Lyons (1991) as moving towards personal goals despite adversity. Connor & Davidson (2003), who also developed an assessment scale in this field after the 2000s, defined resilience as overcoming negative situations. Resilience is more broadly defined by the American Psychological Association as the capacity to manage adversity, trauma, or stressful life events, adapt to the situation, return to regular life more quickly, and subsequently grow and develop (American Psychological Association (APA), 2024; Southwick et al., 2014). Although resilience studies initially began in child development, over the years, an increasing number of academic studies have been conducted for adolescence, adulthood, and old age (Ong, Bergeman & Boker, 2009; Rutter, 1987). Advances in health care and improvements in diagnosis and treatment have increased life expectancy at birth, leading to an increase in the proportion of the world’s population that is ageing. Elderly individuals experience stressful life events such as the loss of a spouse or loved ones, loneliness, financial difficulties due to retirement, death anxiety, and family conflicts, alongside an increasing disease burden and age-related functional limitations as they age (Cosco et al., 2019; Rantanen et al., 2018). The positive adaptation of the elderly exposed to such risks or adverse conditions and developing constructive behaviors to overcome these challenges can be considered psychological resilience (Lima et al., 2023). The combination of longevity and multiple adversities suggests that psychological resilience will become increasingly important for older adults (Seong et al., 2022). In addition, it can be said that among elderly individuals with higher resilience, happiness and life satisfaction increase, depressive symptoms decrease, and accordingly, they undergo a successful ageing process (Li & Ow, 2022). The fact that resilience plays a vital role in the lives of the elderly brings about the need to measure or assess resilience among this population. Most resilience measurement tools published in previous years have been developed for children and adolescents (Oshio et al., 2003; Ungar et al., 2008; Windle, Bennet & Noyes, 2011). However, the difficulties experienced in later life may not directly overlap with the life difficulties of children and adolescents (Fontes & Neri, 2015). Therefore, a specific assessment for older adults is necessary (Cosco et al., 2019). A few measurement tools have also been validated in the elderly population with specific limitations (focusing more on internal resilience and not external factors such as social support) (Cosco et al., 2016; Li & Ow, 2022). In order to address this need, a scale that takes into account the living conditions of the elderly as a whole and utilizes a theoretical model of protective resilience factors (personal protective factors, interpersonal protective factors, spiritual or religious protective factors, experiential protective factors) was developed by Wilson, Plouffe & Saklofske (2022). As far as the researchers have examined the literature, no Turkish resilience scale for older adults is available. The main aim of this study is to adapt the Resilience Scale for Older Adults (RSOA) to Turkish language and culture and to conduct comprehensive validity and reliability analyses. We hypothesise that the adapted Turkish version of the Resilience Scale for Older Adults (RSOA) will demonstrate high reliability, validity and cultural appropriateness for use among older adults in Turkey.

Materials and Methods

This methodologically designed study was conducted between 20 February 2024 and 20 March 2024 with adults over 65 years of age and their relatives (spouses, siblings and close relatives) living in the city centre of Çanakkale and registered at the Altın yıllar Yaşam Merkezi (Golden Years Life Centre). At the beginning of the study, permission was obtained by e-mail from Claire Wilson, the responsible author of the article on the scale to be adapted, to conduct the Turkish validity and reliability study of the scale. Ethical approval was obtained from Çanakkale University Faculty of Medicine Scientific Research Ethics Committee (Approval number: 01/30, Date: 18/01/2024, Number: E-84026528-050.99-2400022858). Permission was also obtained from Çanakkale Municipality to conduct the study in “Altın Yıllar Yaşam Merkezi”.

Sample

In validity and reliability studies, it is recommended that the minimum sample size be at least ten times the total number of items included in the factor analysis (O’Rourke & Hatcher, 2013). The RSOA-T scale, which underwent validity and reliability analysis, comprises 33 items in total. Based on this guideline, the minimum sample size was calculated as 33 × 10 = 330 participants. The purpose of the study and the principles of volunteering were explained to the participants, and their consent was obtained.

The population of the study consists of adults over the age of 65 living in Çanakkale city center and registered at the Altın Yıllar Yaşam Merkezi (Golden Years Life Center), and their spouses, siblings and close relatives. The Altın Yıllar Yaşam Merkezi is a social support centre affiliated to the Municipality of Çanakkale and offers various facilities such as social activities, health services and counselling to individuals over the age of 65. No sample selection was made in the study and it was aimed to reach the entire population. Participants who refused to participate in the study and those with speech or mental communication problems were excluded from the study. A total of 566 volunteers participated in the study.

Participants who provided informed consent were included in the study. The data were collected using face-to-face interview method at Altın Yıllar Yaşam Merkezi, depending on the preferences and availability of the participants. The interview was conducted in a quiet and private area to ensure the comfort and confidentiality of the participants. A questionnaire form created by the researchers was administered. The data collection process took 3 months and was carried out by researchers who ensured consistency in data collection procedures. Face-to-face interview method was used to collect the questionnaire form and it took an average of 25–30 min to complete the questionnaire form.

Language and cultural adaptation

The language and culture adaptation of the scale was carried out following Beaton et al.’s (2000) method, utilizing translation-retranslation and expert opinion. The original items and the final translated scale version were e-mailed to field experts (geriatrics specialists, public health specialists, epidemiologists, and psychiatric nurses) with a good English command for evaluation. The Davis method was used to assess the content validity regarding language and culture (Davis, 1992). The experts were asked to evaluate the items in terms of language and culture by scoring them as ‘inappropriate (1)’, ‘should be seriously revised (2)’, ‘should be slightly revised (3)’, and ‘appropriate (4)’. The content validity index (CVI) was calculated based on the scores for each item. CVI > 0.80 is required. In order to evaluate the comprehensibility of the scale items, a pilot study was conducted with 20 people who fit the study population. These people were excluded from the study.

Data collection tools

The data were collected using a demographic information form developed by the researchers, the Resilience Scale for Older Adults (RSOA), the Perceived Stress Scale (PSS-4), the Older People’s Quality of Life Brief Questionnaire (OPQoL-Brief), and the Geriatric Depression-15 Scale (GDS-15). The authors have permission to use this instrument from the copyright holders.

Sociodemographic form

The researchers’ form included questions about age, gender, education, employment, income, having children, the presence of chronic diseases, medication use, and sleep quality.

Resilience Scale for Older Adults (RSOA)

The RSOA is a scale developed by Wilson, Plouffe & Saklofske (2022) consisting of 33 questions and four sub-dimensions, aiming to measure the resilience and resistance of the elderly. The scale has a four-factor structure: Intrapersonal (12 items), Interpersonal (nine items), Spiritual (six items), and Experiential (six items). The confirmatory factor analysis results of the original scale were χ2(38) = 122.443, p < 0.001, RMSEA = 0.078, CFI = 0.954, TLI = 0.934, SRMR = 0.049, and the factor loadings of the items ranged between 0.40–0.99. Cronbach’s alpha value of the scale ranges between 0.78–0.97. The scale shows positive correlations with life satisfaction and happiness scores and negative correlations with depression, anxiety, and stress scores.

Scoring of the scale items is in the form of “strongly disagree (1)”—“strongly agree (5)”. There are no reverse-scored items in the scale. The dimension item distributions of the scale are ‘Intrapersonal’ (items: 1–12), ‘Interpersonal’ (items: 13–21), ‘Spiritual’ (items: 22–27), and ‘Experiential’ (items: 28–33), and the total score varies between 33–165. An increase in the score indicates an increase in the resilience level of elderly individuals.

Perceived Stress Scale (PSS-4)

The scale developed by Cohen, Kamarck & Mermelstein (1983) was adapted to Turkish language and culture by Eskin et al. (2013). The scale consists of 14 items in total and has two separate short forms of 10 and four items. In this study, the four-item short form was used. Participants rate each item on a five-point Likert-type scale, ranging from ‘Never (0)’ to ‘Very often (4)’. The total score of the questions on the scale varies between 0 and 16. A high score indicates a high level of perceived stress. While the internal consistency coefficient of the scale was 0.66, the test-retest reliability coefficient was calculated as 0.72.

Geriatric Depression-15 Scale (GDS-15)

The first scale developed by Yesavage et al. (1982) consists of 30 questions (Yesavage et al., 1982). A few years later, a short 15-question form was developed for the sick, elderly, and people with dementia (Yesavage & Sheikh, 1986). Later, Burke, Roccaforte & Wengel (1991) conducted a reliability and validity study of the 15-question short form in cognitively intact older adults with mild Alzheimer’s disease. Durmaz et al. (2017) adapted the 15-question short form of the Geriatric Depression Scale to the Turkish language and culture. In this study, a short form with 15 questions was used. While answering the questions, the emotional state over the last week was considered, and the answers consisted of ‘Yes’ and ‘No’ options. While the highest score that can be obtained from the scale is 15, a score of five or above is evaluated in favour of the presence of depression. The internal consistency coefficient of the scale was calculated as 0.92.

Older People’s Quality of Life-Brief Questionnaire (OPQOL-Brief)

The validity and reliability study of the short form of the original 35-item scale was conducted by Bowling et al. (2013). The short form of the scale was adapted to Turkish language and culture in 2019 by Caliskan et al. (2019). The scale, which is in a 5-point Likert format and scored as ‘Strongly Disagree (1)’ and ‘Strongly Agree (5)’, consists of 13 items and one item that is not scored. The total score from the scale varies between 13 and 65, and the higher the score, the higher the quality of life. The internal consistency coefficient of the scale is 0.876, and the test-retest reliability is 0.98.

Statistical analysis

Validity analyses

Exploratory factor analysis (EFA), confirmatory factor analysis (CFA), convergent validity, divergent validity and known-group validity analyses were conducted to evaluate the construct validity of the scale. Principal component analysis was performed in EFA. Factors and eigenvalues were obtained by performing varimax rotation on the items. Items with factor loadings below 0.30 were considered insufficient and the relevant item was removed from the scale when this situation was encountered. In addition, the difference between the factor loadings of the items in different sub-dimensions <0.10 was considered as an overlapping item and the relevant item was removed from the scale when this situation was encountered (Alpar, 2018). CFA is an analysis method that tests whether the sub-dimensions of a predetermined scale are supported. For some fit indices used in CFA, χ2/df < 2, RMSEA < 0.05, NFI > 0.95, CFI > 0.95, RFI > 0.95 and IFI > 0.95 values indicate a good fit; χ2/df < 5, RMSEA < 0. 08, NFI > 0.90, CFI > 0.90, RFI > 0.90 and IFI > 0.90 are accepted as acceptable fit limits (Schermelleh-Engel, Moosbrugger & Müller, 2003). Convergent validity assessment was conducted on the scales that were expected to show a positive correlation with the scale whose validity was tested, and divergent validity assessment was conducted on the scales that were expected to show a negative correlation. These assessments were performed using Pearson correlation test. Known-group validity was tested by evaluating the relationship between independent variables and RSOA-T scale scores.

Reliability analyses

Cronbach’s alpha was calculated to evaluate the scale’s internal reliability. A Cronbach’s alpha of 0.70 and above was determined as the acceptable limit. The split-half method was used to evaluate scale reliability. The whole scale and sub-dimensions were randomly divided into two halves using the SPSS 23.0 package program. Spearman-Brown and Guttman’s reliability coefficients were evaluated for the randomly divided halves. Spearman-Brown and Guttman’s reliability coefficients of 0.70 and above are considered acceptable limits (Alpar, 2018).

Data analyses

Categorical variables were presented as numbers and percentages, while continuous variables were presented as mean (x¯), standard deviation (S), minimum (min.), and maximum (max.). The normal distribution of the data was evaluated using analytical methods (Kolmogorov-Smirnov), histograms, and probability graphs. The correlation between continuous variables was evaluated using the Pearson correlation test. The absolute value of the correlation coefficient was interpreted as a weak correlation if r < 0.30, a moderate correlation if 0.30–0.50, and a strong correlation if r > 0.50 (Cohen, 1988). Differences between independent groups were analyzed using the Student’s t-test for two group evaluations and the ANOVA test for three or more group evaluations. IBM SPSS 23.0 and LISREL 8.80 package programs were used to evaluate the research data statistically. This study set the statistical significance level at p < 0.05.

Results

Sociodemographic characteristics

The mean age of the 566 elderly individuals participating in the study was 71.7 ± 5.2 years (range: 65–92), and 56.2% (n = 318) were women. The sociodemographic and health-related characteristics of the elderly are shown in Table 1.

Table 1 Sociodemographic and health characteristics of the study population.

Features	n (%)	
Education		
Primary school	212 (37.5)	
Middle school	64 (11.3)	
High school	122 (21.6)	
University	168 (29.7)	
Marital status		
Married	345 (61.0)	
Widowed	171 (30.2)	
Single	50 (8.8)	
Cohabitant		
Relative or spouse	340 (60.1)	
Alone	226 (39.9)	
Income perception		
My income is more than my expenditure	131 (23.1)	
My income is equal to my expenditure	373 (65.9)	
My income is less than my expenditure	62 (11.0)	
Chronic illness		
Present	444 (78.4)	
Absent	122 (21.6)	
Sleep quality		
Good	208 (36.7)	
Average	273 (48.2)	
Poor	85 (15.0)	

Validity

Content validity

Evaluations obtained from nine experts (geriatrics specialist, psychiatrist, public health specialist, psychiatric nurse), consisting of experts and academicians working in the field of elderly health or mental health, showed that the CVI of the items was between 0.88 and 1.00, and the CVI of the scale was 0.98. The scale has a CVI above 0.80 and is at a reasonable level.

Exploratory factor analysis (EFA)

The exploratory factor analysis determined that the scale items loaded onto four dimensions with varimax rotation. As a result of the exploratory factor analysis of thirty-three items, the Kaiser-Meyer-Olkin Measure was found to be 0.931 (p < 0.001), and the Bartlett sphericity test value was x2 =14,978.5 (df = 528; p < 0.001). The four dimensions of the scale account for 62.61% of the total variance. The distribution of factor loadings of the scale items: The factor loading of factor 1 varies between 0.406–0.760, factor 2 between 0.908–0.947, factor 3 between 0.502–0.827, and factor 4 between 0.509–0.808 (Table 2).

Table 2 Factor loadings, item distribution, item-total correlation, and Cronbach’s alpha of the Resilience Scale for Older Adults-Turkish.

Items	F1	F2	F3	F4	x¯ ± S	Item-total correlation	Cronbach’s alpha if item deleted	
I1	0.736	–	–	–	4.18 ± 0.86	0.552	0.931	
I2	0.760	–	–	–	4.23 ± 0.85	0.587	0.931	
I3	0.743	–	–	–	4.19 ± 0.94	0.555	0.931	
I4	0.659	–	–	0.357	4.39 ± 0.79	0.640	0.931	
I5	0.739	–	–	–	4.18 ± 0.86	0.628	0.931	
I6	0.579	–	–	0.344	4.34 ± 0.81	0.542	0.931	
I7	0.653	–	–	–	4.07 ± 1.01	0.597	0.931	
I8	0.616	–	–	–	4.10 ± 0.87	0.553	0.931	
I9	0.703	–	–	–	4.07 ± 0.96	0.587	0.931	
I10	0.533	–	0.306	0.382	4.40 ± 0.78	0.590	0.931	
I11	0.642	–	–	–	4.08 ± 0.97	0.495	0.932	
I12	0.406	–	–	–	3.49 ± 1.25	0.311	0.933	
I13	–	–	0.573	0.399	4.45 ± 0.84	0.587	0.931	
I14	0.331	–	0.606	–	4.34 ± 0.92	0.588	0.931	
I15	–	–	0.652	–	4.28 ± 0.95	0.581	0.931	
I16	–	–	0.827	0.302	4.48 ± 0.88	0.592	0.931	
I17	–	–	0.822	–	4.45 ± 0.93	0.568	0.931	
I18	–	–	0.823	–	4.47 ± 0.91	0.586	0.931	
I19	0.390	–	0.606	–	4.08 ± 1.04	0.548	0.931	
I20	0.384	–	0.625	–	3.99 ± 1.08	0.555	0.931	
I21	–	–	0.502	–	3.75 ± 1.25	0.439	0.933	
I22	–	0.908	–	–	4.11 ± 1.22	0.550	0.931	
I23	–	0.947	–	–	4.03 ± 1.25	0.551	0.931	
I24	–	0.914	–	–	3.87 ± 1.37	0.500	0.932	
I25	–	0.946	–	–	4.07 ± 1.25	0.523	0.932	
I26	–	0.942	–	–	4.12 ± 1.22	0.532	0.932	
I27	–	0.928	–	–	3.98 ± 1.27	0.518	0.932	
I28	–	–	–	0.757	4.58 ± 0.64	0.524	0.932	
I29	–	–	–	0.743	4.51 ± 0.78	0.384	0.933	
I30	–	–	–	0.808	4.56 ± 0.70	0.514	0.932	
I31	0.348	–	–	0.650	4.63 ± 0.65	0.588	0.931	
I32	0.344	–	–	0.509	4.32 ± 0.77	0.471	0.932	
I33	0.465	–	–	0.589	4.47 ± 0.72	0.614	0.931	
Variance explained (%)	19.42	16.45	14.65	12.08	Total = 62.61			
Note:

Factor Loadings >0.30 Indicated.

Confirmatory factor analysis (CFA)

It was determined that the Resilience Scale for Older Adults-T (RSOA-T) consists of four dimensions with the same structure as the original scale, which we had previously determined with EFA. With the application of the three proposed modification indices, it was determined that the fit index values achieved good or acceptable values as a result of CFA (Table 3). The RSOA-T, consisting of thirty-three items and four dimensions, contains no item with a factor load below 0.30 due to the CFA analysis. There are 12 items in the Intrapersonal (F1) dimension. The factor load varies between 0.33–0.80, with nine items in the Interpersonal (F2) dimension with factor loadings ranging from 0.42–0.84, six items in the Spiritual (F3) dimension with factor loadings between 0.90–0.95, and six items in the Experiential (F4) dimension with factor loadings between 0.54–0.80 (Fig. 1). The scale consists of 33 items in total and is scored between 33 and 165. The scale scoring across dimensions is as follows: “Intrapersonal dimension” ranges from 12 to 60, “ Interpersonal dimension” from 9 to 45, “Spiritual dimension” from 6 to 30, and the “Experiential dimension” from 6 to 30.

Table 3 Confirmatory factor analysis fit indices of the Resilience Scale for Older Adults-Turkish.

Index	RSOA-T	Good fit	Acceptable fit	Model fit	
χ2/df	4.24	<2	<5	Acceptable fit	
RMSEA	0.076	<0.05	<0.08	Acceptable fit	
NFI	0.95	>0.95	>0.90	Acceptable fit	
CFI	0.96	>0.95	>0.90	Good fit	
RFI	0.95	>0.95	>0.90	Acceptable fit	
IFI	0.96	>0.95	>0.90	Good fit	
Note:

RSOA-T, Resilience Scale for Older Adults-Turkish.

Figure 1 Factor loading diagram of confirmatory factor analysis results for the Resilience Scale for Older Adults-Turkish.

F1: Intrapersonal, F2: Interpersonal, F3: Spiritual, F4: Experiential.

Determination of scale discrimination and construct validity with alternative scales

The mean RSOA-T score of 566 participants was calculated as 139.3 ± 18.2 and the scores ranged between 33 and 165. In terms of the relationship between the sociodemographic characteristics of the participants and the RSOA-T total score, it was found that the mean scale score of women (141.3 ± 17.7) was statistically significantly higher than that of men (136.7 ± 18.4) (p = 0.003). The correlation between age and RSOA-T total score was weak (r = 0.100) and statistically significant (p = 0.017). Regarding educational status, the mean scale score of those with high school and lower education levels (141.4 ± 17.7) was found to be higher than those with university levels (134.3 ± 18.5) (p < 0.001). Regarding marital status, the scale scores of single people (132.3 ± 15.9) were statistically significantly lower than those who were married (139.5 ± 18.8) and widowed (140.9 ± 17.4) (p = 0.024, p = 0.008, respectively). The mean RSOA-T score was significantly lower in those whose income was less than their expenses (132.5 ± 20.5), whose income was equal to their expenses (140.1 ± 18.4) and whose income was more than their expenses (140.3 ± 16.1) (p = 0.007, p = 0.016, respectively). There was no statistically significant difference between having a chronic disease and the mean RSOA-T score (p = 0.402). It was found that the mean scores of those with good sleep quality (143.7 ± 14.2) were statistically higher than those with moderate (137.0 ± 19.2) and poor (135.8 ± 21.5) sleep quality (p < 0.001, p < 0.001, respectively).

Three scales with convergent and divergent features were applied to the RSOA-T. A strong positive relationship (r = 0.657) was found with the OPQOL-Brief, which is expected to be convergent with the RSOA-T scale (p < 0.001). When the relationship with PSS-4 and GDS-15, which are divergent from the RSOA-T scale, was evaluated, a negative moderate correlation (r = −0.330) with PSS-4 and a negative moderate correlation (r = −0.338) with GDS-15 was found (p < 0.001, p < 0.001, respectively) (Table 4).

Table 4 Correlation analysis of the Resilience Scale for Older Adults-Turkish and its dimensions with convergent and divergent scales.

Scale		PSS-4	OPQOL-Brief	GDS-15	
RSOA-T	r (p)	−0.330 (<0.001)	0.657 (<0.001)	−0.338 (<0.001)	
Intrapersonal	r (p)	−0.418 (<0.001)	0.675 (<0.001)	−0.455 (<0.001)	
Interpersonal	r (p)	−0.276 (<0.001)	0.608 (<0.001)	−0.333 (<0.001)	
Spiritual	r (p)	−0.04 (0.341)	0.138 (0.001)	−0.008 (0.855)	
Experiential	r (p)	−0.219 (<0.001)	0.562 (<0.001)	−0.135 (0.001)	
Note:

RSOA-T, Resilience Scale for Older Adults-Turkish; PSS-4, Perceived Stress Scale; OPQOL-Brief, Older People’s Quality of Life-Brief; GDS-15, Geriatric Depression Scale-15.

Reliability

Cronbach’s alpha internal consistency coefficient calculated for the whole scale was 0.933, which varies between 0.862 and 0.975 for the sub-dimensions (Table 5). The scale total score correlation coefficient of the scale items ranges between 0.311 and 0.640 (Table 2). According to the split-half analysis, the Spearman-Brown coefficient for the whole scale was 0.723, ranging between 0.752 and 0.964 for its sub-dimensions. Guttman Split-half values were 0.723 for the whole scale and 0.752–0.963 for its sub-dimensions (Table 5).

Table 5 Reliability analysis of the Resilience Scale for Older Adults-Turkish.

Scale and dimensions	Cronbach alpha	Spearman-Brown	Guttman split-half	x¯ (S)	Min.-Max.	
RSOA-T	0.933	0.723	0.723	139.3 (18.2)	33–165	
Intrapersonal	0.904	0.809	0.809	49.7 (7.7)	12–60	
Interpersonal	0.892	0.787	0.786	38.3 (6.5)	9–45	
Spiritual	0.975	0.964	0.963	24.2 (7.2)	6–30	
Experiential	0.862	0.752	0.752	27.1 (3.3)	6–30	
Note:

RSOA-T, Resilience Scale for Older Adults-Turkish.

Discussion

In light of the growing elderly population, it is of great importance for healthcare professionals to accurately assess psychological resilience in older adults. The RSOA-T adapted into Turkish, is a relatively practical tool. A comprehensive assessment is also provided by the four-factor structure (intrapersonal protective, interpersonal protective, spiritual, and experiential protective factors). The scale allows for scoring factors that predict protective factors for resilience while also allowing for calculating an overall total score (Wilson, Plouffe & Saklofske, 2022). A distinctive aspect of the scale is its theoretical foundation in qualitative research, which sets it apart from other scales in this field. While the existing scales in this field are based on quantitative studies and researchers’ conceptualisations, this scale was developed from a model grounded in interviews conducted as part of qualitative research with older adults (Connor & Davidson, 2003; Martin et al., 2015). The model developed from older adults’ perspectives on resilience and the resulting scale offers a more appropriate and meaningful framework for defining resilience in the elderly (Wilson, Walker & Saklofske, 2021). The analyses revealed that the scale, adapted to the Turkish language and culture, possesses strong construct validity and reliability.

In the adapted RSOA-T scale, it was found that female participants demonstrated higher levels of resilience compared to male participants in terms of total score (p < 0.003). The results of studies in this field are variable. For example, a study investigating multidimensionally resilience during the COVID-19 pandemic found that women had better perceived health and higher resilience levels (p < 0.01). Similarly, in the scale development study by Perez-Rojo et al. (2022), Martin et al. (2015) for measuring individual and interpersonal resilience, women had higher resilience levels (p < 0.05). Nevertheless, in a separate resilience scale development study, no statistically significant difference was observed between the resilience scores of female and male participants (p < 0.520) (von Eisenhart Rothe et al., 2013). The study conducted by Martínez-Moreno et al. (2020) examined the effects of gender, age, and physical activity on resilience. The findings indicated that men exhibited higher levels of resilience than women. Marital status is another variable for which significant differences in resilience scores were observed. Single individuals’ resilience scores were lower than those of widowed and married individuals. Furthermore, numerous studies have demonstrated that single individuals exhibit lower levels of resilience than their married counterparts (Martínez-Moreno et al., 2020; Martin et al., 2015; Parmaksız, 2020). The enhanced resilience observed in married individuals may be attributed to their more straightforward access to spousal or social support during adverse life circumstances (Ang et al., 2018). The findings of our study indicated a significant correlation between good sleep quality and high resilience scores (p < 0.001). It has been demonstrated in previous studies that enhanced psychological resilience can prevent a decline in sleep quality, even when individuals are confronted with stressful life events (Li et al., 2019). More precisely, resilience functions to mitigate the effects of stress that have a deleterious impact on sleep (Liu et al., 2016). An increasing body of evidence from scientific studies has demonstrated the regulatory function of resilience in sleep quality (Du et al., 2020; Lenzo et al., 2022).

Discussion of validity analysis

Given that the content validity index of the scale items ranges from 0.88 to 1.00, it can be interpreted that the conceptual structure that the scale aims to measure is primarily represented. The scale’s KMO coefficient was found to be 0.931, sufficient for evaluating the sample’s factor structure (Alpar, 2018). The results of the exploratory factor analysis indicated the presence of four factors, which collectively explained 62.61% of the total variance. The explained variance rate is recommended to exceed 50% of the total variance. Therefore, it can be stated that the four-factor structure of the scale is an appropriate representation of the scale, given that it accounts for 62.61% of the total variance (Yaşlıoğlu, 2017). Upon examination of the factor loadings, it can be stated that only one of the items under factor 1 (Item 12: I try to live each day as if it were my last) will be classified as poor, with a value of 0.406. However, the other items’ factor loadings will be within the normal, good, and excellent categories (O’Rourke & Hatcher, 2013). An examination of the goodness of fit indices for confirmatory factor analysis revealed that the comparative goodness of fit index, IFI and CFI, showed good fit. Furthermore, all item-total correlations of the scale exceeded the acceptable value of 0.30. The item-total correlations ranged from 0.311 (Item 12) to 0.640 (Item 4: When I put my mind to a task, I can effectively complete it). In the original scale RSOA, item loadings in the CFA analysis ranged between 0.418 and 0.995 (Wilson, Plouffe & Saklofske, 2022). In the scale we adapted, Item 12 is above the acceptable limit of 0.30 but loads less than the other items in both EFA and CFA. This situation led us to consider that, besides the language barrier, the participants’ cultural perspectives and conceptual understandings of this item may differ. The idea that “I live every day as if it were my last day,” as stated in Item 12, prioritises living in the moment and focusing on the present without worrying about the future or dwelling on past experiences. Nevertheless, the dominant perspective in our society posits an afterlife characterised by eternal peace and happiness, with the tangible benefits and attractions of the physical world in which we live taking a secondary position. Therefore, this item may not have elicited a significant response from the participants (Joshanloo, 2013). In order to test the equivalence criteria of RSOA-T, the PSS-4, GDS-15 and OPQOL-Brief were employed (Eskin et al., 2013; Durmaz et al., 2017; Caliskan et al., 2019). The strongest correlation between these scales was found with the OPQOL-Brief scale (r = 0.657) (p < 0.001). In the case of the original RSOA scale, it was established that higher scores on its sub-dimensions and, in total, were associated with an enhanced quality of life (Wilson, Plouffe & Saklofske, 2022). The results of our study are by those of other studies, which demonstrate that quality of life and resilience are closely interrelated and that quality of life resilience is typically expressed as the outcome variable (Hicks & Conner, 2014; Sihvola, Kuosmanen & Kvist, 2022). Quality of life is a multidimensional concept that evaluates individuals’ general well-being, satisfaction and satisfaction with life. It also includes health status in the context of life (Chuang, Wu & Wang, 2023). Studies have shown that people with a high level of resilience have a better quality of life (Post et al., 2018; Wartelsteiner et al., 2016). In the study conducted by Las Hayas et al. (2016) with patients, it was found that resilience contributed to improving patients’ quality of life and accelerated the recovery process. The study by Calvete, Las Hayas & Gómez Del Barrio (2018) found that the relationship between resilience and quality of life was reciprocal and that resilience factors led to an increase in several domains of quality of life.

Our study found a negative and moderately significant correlation between the psychological resilience scale and the perceived stress scale (r = −0.330, p < 0.001). In this case, resisting the negative effects of stressors and preventing the emergence of serious dysfunctions that may occur are also related to the resilience capacity of the individual (Babić et al., 2020). In other words, individuals with high resilience have better quality of life and less impairment in daily life activities when exposed to stress compared to other individuals (Bhatnagar, 2021). Similar to our study, partial correlation results between the Brief Resilience Scale developed by Smith et al. (2008) and the Perceived Stress Scale were found as r = −0.38 p < 0.001 for undergraduate students and r = −0.46 p < 0.01 for cardiac patients.

Depression in older adults can be defined as a debilitating condition in the long term, with less remission and a tendency to recur more frequently than in younger people. Within the framework of the aetiology of depressive disorder, psychological resilience appears as a factor that reduces the onset or recurrence risk of depression, positively affects the reduction of disease severity and increases the rate of recovery. In other words, psychological resilience minimises an individual’s lifelong risk of depression (Laird et al., 2019). In our study, a significant negative correlation was found between the geriatric depression scale and the psychological resilience scale (r = −0.338, p < 0.001).

According to the results of the factor analyses, it is seen that the highest loading items of the scale are in the Spiritual sub-dimension. The items in this sub-dimension include concepts such as asking for help through prayer, believing that God protects and trusting in God. Religiosity or spirituality is generally explained on the basis of belief in a supernatural power. Religion or spirituality can provide a reassurance against loneliness by making people feel the presence of God and in this context, it can help individuals cope with difficulties (Wilson, Walker & Saklofske, 2021). According to Carl Gustav Jung, the state of belief or worship creates a basis for the unconscious to constantly remember prototypes and associate them with consciousness. According to Jung, in this process, formulae and images enable the unconscious to express its instinctive movements sufficiently and to transfer these expressions to the consciousness smoothly. In this way, consciousness never loses its instinctive roots. In this context, Jung stated that he considered religions as a comprehensive psychotherapy system (Clark, 2018). The majority of older adults living in Turkey are Muslims. The Qur’an, the holy book of Muslims, emphasises that inner peace can be achieved by remembering God, praying and asking for His help. According to the Islamic belief in Turkey, religious individuals think that Allah, the greatest power in the universe, protects and supports them. They also believe that everything that happens to them is within God’s knowledge, that they are not left alone, and that there is a God close to them (Lameei & Bilici, 2021). In a study conducted on adults in Turkey, the relationship between religiosity and psychological resilience was examined and it was found that as the scores obtained from the religiosity scale increased, psychological resilience also increased and there was a significant positive correlation between the two variables (Yıldırım & Gürsu, 2021). These findings suggest that lifestyles in Turkish-Islamic culture may find more correspondence in the spiritual sub-dimension of the scale.

It is seen that the items in the interpersonal sub-dimension, which is one of the sub-dimensions of the scale, have higher item factor loadings than those in the intrapersonal sub-dimension. The interpersonal sub-dimension refers to the family, friends, neighbours or communities that older adults can get support from when they need it. When we consider this situation in the context of Turkish culture, it is seen that behaviours such as respecting parents, listening to their words and advice, meeting their needs gratuitously and undertaking their health care when necessary are still maintained as a traditional approach (Mandıracıoğlu, Lüleci & Özvurmaz, 2017). Through this situation, older adults feel valuable and do not feel lonely. The concept of neighbourhood has a special importance in Turkish society. Making a welcome visit to a neighbour who has just moved in, having children play together if they are close in age, taking food to a neighbour who is sick, and being together at events such as weddings or deaths are traditional customs that still continue. Such social support mechanisms positively affect and strengthen the psychological resilience of individuals (Gündüz & Yıldız, 2008). Turkey, due to its geographical location, tends to exhibit a more collectivist approach compared to Western states. Although this situation is not absolute, it is seen that individuals show less individual characteristics, have stronger social ties, and values such as material and moral solidarity and empathy are more common. Such social characteristics may support psychological resilience by contributing to the development of a sense of trust and commitment in older adults (Özyer, Orhan & Dönmez Orhan, 2012).

Discussion of reliability analysis

The Cronbach alpha coefficient for RSOA-T was determined to be 0.933, while the sub-dimensions exhibited coefficients between 0.862 and 0.975. This value demonstrates that the scale is highly reliable (Alpar, 2018). The Cronbach alpha coefficients of the sub-dimensions of the original scale (RSOA) have been found to vary between 0.82 and 0.98 (Wilson, Plouffe & Saklofske, 2022). An examination of other scales related to resilience reveals that the Cronbach’s alpha values of the Connor-Davidson Resilience Scale (CD-RISC), one of the most popular scales for use with the elderly, range between 0.88 and 0.93 (Goins, Gregg & Fiske, 2013). The Cronbach’s alpha coefficients of the Resilience Scale 11 (RS-11) and Resilience Scale 5, also employed to assess resilience in the elderly, are 0.86 and 0.80, respectively (von Eisenhart Rothe et al., 2013). The internal consistency coefficient of the resilience scale for the elderly, developed by Li & Ow (2022), is relatively novel compared to other scales, with a value of 0.88. A comparison of the internal consistency coefficient of the adapted scale, RSOA-T, with other studies indicates that it is at a similar level and reliable.

When the Cronbach alpha reliability coefficient value of the RSOA-T was examined in studies developed or adapted in Turkish culture, it was found that the Cronbach alpha coefficient determined for both the sub-dimensions and the whole scale was high according to them. Examples of studies conducted in Turkey were given. In the study ‘The Reliability and Validity of the Resilience Scale for Adults Turkish Version’ conducted by Basım & Çetin (2011) in adults aged 18–37 years, the internal consistency coefficients of the sub-dimensions of the scale were found to be between 0.66 and 0.81. While the Cronbach alpha reliability coefficient for the entire ‘The Psychological Hardiness Scale’ developed by Işık (2016) in adults aged 20–47 years was 0.76, the Cronbach alpha reliability coefficients for each sub-dimension were between 0.62 and 0.74. Again, the cronbach alpha value of the scale developed to measure the psychological resilience levels of early adolescents in Turkey was found to be 0.85 for the whole scale and between 0.72 and 0.78 for its sub-dimensions (Baltaci & Karataş, 2014). According to these studies, the fact that the RSOA-T scale we adapted has high internal consistency levels shows us that the items in the scale are consistent with each other.

Limitations

It should be noted that the present study is not without limitations. The majority of participants were urban residents who had completed secondary education or higher and had a relatively high income. A positive effect on resilience was observed in participants who had a good education and income, and who resided in an urban area with convenient access to healthcare facilities and government institutions. It is, therefore, possible that the resilience scores of the participants were affected by this. Including individuals residing in rural areas with low education and income levels in the study was not feasible. Furthermore, the participants in the study sample were afflicted with chronic diseases. Yet, they did not suffer from severe health issues that would negatively impact or restrict their quality of life. This is because these individuals regularly engaged in various activities (including exercise, choir activities, painting, and so forth) at the Golden Years Healthy Living Centre. In this case, individuals actively struggling with serious health problems (e.g., hospitalised or treated at home) were excluded from the sample. Another limitation of the study sample is that the participants were limited to individuals residing in Çanakkale. As in every country, there are various socio-cultural and economic differences between provinces in Turkey. The inability to reach participants from different regions also constitutes a limitation of the sample. Secondly, due to the original scale structure, it can be posited that this scale is focused on the protective factors of resilience. In other words, the scale did not assess the potential risks or vulnerabilities that may be effective in the endurance process. To address this limitation and gain insight, we examined the relationship between endurance and various variables, including gender, sleep, etc. In this context, we conducted an evaluation of the relationship between sleep quality and endurance using a single question. The use of a single question and a self-reported score may have reduced the reliability of our results, although similar studies in the literature exist. A further limitation of the study is that the participants were not asked about stressful events that they had previously encountered or that were currently affecting them. In addition, it is our understanding that this is the first resilience scale for the elderly in Türkiye. We have completed the adaptation process with the utmost precision, ensuring that no items or dimensions were lost. Another limitation of the study is that the data on resilience in the elderly were collected only through questionnaires and scales. In self-report scales, data may be incomplete or inaccurate due to misremembering, misunderstanding or differences in interpretation. In addition, the fact that different data collection methods were not included in this study (e.g., in-depth interviews, focus group discussions, observation) may limit the depth and scope of the study.

Conclusions

The increasing age of the population has brought about a period of heightened concern for the physical, mental and social health of older individuals. The process of developing and maintaining resilience in individuals is of critical importance in ensuring positive adaptation to challenging and stressful life events, as well as in facilitating successful ageing. The RSOA-T is a comprehensive, well-constructed, highly valid and reliable scale for older adults. The scale may be employed as a fundamental instrument for comprehending older adults’ resilience levels. It would be beneficial for future studies to consider testing the RSOA-T in a clinical sample. As psychological resilience significantly impacts both the psychosocial and physical health of older people, this scale can be an valuable assessment tool in multidisciplinary teams (e.g., psychiatrists, geriatricians, social workers). The scale can be utilized to identify individuals with low resilience and to understand whether they require additional clinical support. In particular, when situations such as loneliness, low social support or lack of spiritual resources are identified, interventions can be made for these individuals. As psychological resilience determines the capacity of individuals to cope with stressful life events and traumas, this scale can reveal the risk of mental disorders such as depression and anxiety in older individuals in a more detailed way when used with other scales (depression and anxiety scales, etc.). An alternative approach would be to develop a culturally appropriate scale using data from a qualitative study that comprehensively examines resilience in older adults. It is recommended that studies on resilience in older adults be conducted over a more extended period of time and include risk factors, with particular attention paid to those affecting the Turkish population. Preferring mixed methods (using qualitative and quantitative data collection methods together) in future studies on resilience will enrich the findings on resilience and develop a more comprehensive understanding. Although the quality of life scale was used for the convergent scale validity of the resilience scale in older adults in this study, looking at the relationship of the scale with other constructs such as coping strategy or social support related to resilience in future studies will strengthen the use of the scale. Once the risk factors, protective factors and various psychopathologies have been identified, intervention and prevention programmes should be devised and implemented.

Supplemental Information

Supplemental Information 1 Questionnaire (English).

Supplemental Information 2 Questionnaire (Turkish).

Supplemental Information 3 Survey dataset.

We would like to express our sincere thanks to academicians and experts Aliye Mandıracıoğlu, Aslı Davas Erdemli, Bedriye İlkay Aygül, Coşkun Bakar, Gamze Çan, Hilal Heybeli, Leyla Baysan Arabacı, Meltem Çiçeklioğlu and Sibel Oymak for their opinions on the content validity of the scale.

Additional Information and Declarations

Competing Interests

Author Contributions

Human Ethics

Data Availability

The authors declare that they have no competing interests.

Seher Palanbek Yavaş conceived and designed the experiments, performed the experiments, analyzed the data, prepared figures and/or tables, authored or reviewed drafts of the article, and approved the final draft.

Caner Baysan conceived and designed the experiments, performed the experiments, analyzed the data, prepared figures and/or tables, authored or reviewed drafts of the article, and approved the final draft.

The following information was supplied relating to ethical approvals (i.e., approving body and any reference numbers):

Ethical approval was obtained from Çanakkale University Faculty of Medicine Scientific Research Ethics Committee (Approval number: 01/30, Date: 18/01/2024, Number: E-84026528-050.99-2400022858).

The following information was supplied regarding data availability:

The survey data is available in the Supplemental File.

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
