# Peer review of "Turkish validation of a resilience scale from older people’s perspectives: resilience scale for older adults"

_PeerJ, doi:10.7717/peerj.18837_

## Round 0.1 · original submission · Major Revisions

The manuscript presents the adaptation and validation of the Resilience Scale for older adults in the Turkish context, employing a rigorous methodological framework that includes exploratory and confirmatory factor analyses, a high content validity index, and robust reliability measures. While the study demonstrates strong methodological foundations and offers a valuable contribution to resilience research, several areas require refinement to enhance clarity, coherence, and impact.

Both reviewers acknowledged the strengths of the manuscript, particularly its methodological rigor. Reviewer 1 highlighted the effective use of exploratory and confirmatory factor analyses to assess construct validity and the robust reliability measures. These strengths are further supported by the demonstration of convergent and discriminant validity. Reviewer 2 also recognized the study's relevance to resilience research but emphasized the need for a stronger alignment between the title, objectives, and findings. The current title does not fully reflect the scope of the study, which explores the relationships between resilience and factors such as quality of life, stress, and depression. Revising the title to better represent the study's focus would improve its clarity.

In terms of basic reporting, both reviewers identified areas needing improvement. Reviewer 2 pointed out that the main objectives and hypotheses of the study are not clearly articulated in the abstract or introduction, making it difficult for readers to grasp the central focus. Reviewer 1 suggested providing a richer description of the sample, including demographic details such as education level and urban or rural residence, to improve the generalizability of the findings. Additionally, Reviewer 2 recommended that the manuscript clearly identify the knowledge gap it seeks to address and provide evidence of factors influencing resilience in older adults.

The experimental design, while robust, requires additional detail and justification to enhance transparency and methodological rigor. Reviewer 1 suggested elaborating on the sample size calculation and the rationale for using specific factor analyses. Reviewer 2 raised concerns about participant recruitment, research settings, and ethical considerations, noting that these aspects were not adequately described. Furthermore, both reviewers emphasized the importance of discussing the study's limitations, such as potential sampling biases and reliance on self-reported data, which may introduce response bias. Reviewer 1 proposed incorporating a multi-method approach, such as interviews or behavioral observations, to provide a more comprehensive understanding of resilience.

The validity of the findings is supported by the rigorous analyses and strong reliability measures, as noted by Reviewer 1. However, both reviewers identified areas for further exploration. Reviewer 1 suggested that the study could be strengthened by investigating additional constructs, such as coping strategies and social support, to enhance the scale's utility. Reviewer 2 emphasized the need to align the results and discussion more closely with the study's stated objectives and hypotheses. Both reviewers also recommended addressing cultural nuances that may influence resilience in the Turkish context, as this would provide deeper insights and greater contextual relevance.

Finally, both reviewers emphasized the need for a clearer discussion of the study's clinical and practical implications. Reviewer 2 specifically suggested elaborating on how the findings could inform interventions aimed at improving resilience among older adults, particularly in relation to quality of life, stress, and depression.

The manuscript has potential but requires revisions to improve its clarity, coherence, and impact. The title and abstract should be refined to better reflect the study’s objectives, and the main focus should be articulated more clearly. The methodological section would benefit from a more detailed description of participant recruitment, ethical considerations, and the rationale for the analyses used. The discussion should address the study's limitations and the cultural factors influencing resilience, providing a more nuanced interpretation of the findings. Aligning the results and discussion with the stated objectives and hypotheses is critical, as is expanding on the clinical implications of the study. With these revisions, the manuscript will offer a valuable contribution to resilience research and its practical applications in the Turkish context.

·

Basic reporting

The manuscript provides a thorough methodological approach to adapting the resilience scale for older adults to the Turkish context. The use of both exploratory and confirmatory factor analyzes to assess construct validity is a strength, as are the reported high content validity index and robust reliability measures. More detailed reporting on the sample characteristics, such as: However, adding socio-demographic factors (e.g. level of education, urban or rural residence) would improve the generalizability of the results. The manuscript could also benefit from a clearer discussion of limitations, particularly with regard to possible sampling biases. Further research into the cultural nuances of resilience would be valuable. With minor revisions it would be suitable for publication in PeerJ.

Experimental design

The study's experimental design is generally robust, but some improvements could improve its clarity and accuracy. First, while the inclusion of factor analyzes is appropriate for construct validity, more detailed information on how to calculate sample size and justify the use of these specific analyzes would strengthen methodological rigor. Additionally, the study relies on self-reported data from scales, which may introduce response bias; Incorporating a multi-method approach (e.g. interviews or behavioral observations) could provide richer data. Finally, the study could address potential cultural factors affecting resilience, which could help interpret the results more thoroughly in the Turkish context.

Validity of the findings

The manuscript provides a thorough validation of the resilience scale for older adults in the Turkish context and demonstrates strong construct, convergent and discriminant validity. The factor structure is preserved and the reliability is robust. However, the sample lacks diversity in terms of demographic characteristics such as socioeconomic status or rural-urban living, which may limit the generalizability of the results. Furthermore, while convergent validity with the quality of life scale has been demonstrated, further examination of other related constructs (e.g., coping strategies, social support) would strengthen the usefulness of the scale. Future studies could examine test-retest reliability to further validate it over time.

Additional comments

This study effectively presents the Turkish validation of the Resilience Scale for older adults and demonstrates its reliability and validity for assessing resilience in this population. Methodological rigor, including factor analyzes and various reliability assessments, underpins the results. However, further clarification of sample selection and possible cultural influences on resilience could improve the contextualization of the results. Furthermore, future research could examine the applicability of the scale in different Turkish subgroups or with longitudinal designs to better understand resilience over time. Overall, the study provides valuable insights, although a deeper examination of cultural relevance would strengthen the conclusions.

Reviewer 2 ·

Basic reporting

1. Your title, it’s seemed that you look for the validity of a resilience scale among older persons. However, your results revealed about relationship between resilience and quality of like, stress, and depression. Please revise.
2. Abstract in this article should be unstructured or structure abstract?
3. What is main topic that this article wants to focus? It’s not clear.
4. Please identify clearly objective and hypothesis of the study.
5. If you focused on the resilience of older adults, you might have to show level of resilience in older adults and how it is problem.
6. Then, provide evidence about factors related to the resilience among the older adults. Especially, quality of life, stress, and depression.
7. Please provide gap of knowledge of the study.

Experimental design

8. How is the authors recruit the participants?
9. How is data collection? Where are research settings?
10. Please mention ethical consideration in this study?
11. what are limitations of this study?

Validity of the findings

12. Results; please present based on your objective and hypotheses.
13. Discussion also depend on your objective and hypotheses.

Additional comments

14. please explain more clinical implication

Annotated reviews are not available for download in order to protect the identity of reviewers who chose to remain anonymous.

---

## Round 0.2 · accepted · Accept

Thank you for submitting the revised version of your manuscript. After carefully reviewing the changes, I can confirm that all reviewers' comments and suggestions have been appropriately addressed.

Based on this assessment, I am pleased to inform you that the manuscript is now ready for publication. Congratulations.

·

Basic reporting

The manuscript requires no furtenr modifications.

Experimental design

The manuscript requires no furtenr modifications.

Validity of the findings

The manuscript requires no furtenr modifications.

Additional comments

The manuscript requires no furtenr modifications.

Reviewer 2 ·

Basic reporting

It's clear writing.

Experimental design

after revised, it's well done.

Validity of the findings

ok